# Lentil Fortification and Non-Conventional Yeasts as Strategy to Enhance Functionality and Aroma Profile of Craft Beer

**DOI:** 10.3390/foods11182787

**Published:** 2022-09-09

**Authors:** Laura Canonico, Alice Agarbati, Emanuele Zannini, Maurizio Ciani, Francesca Comitini

**Affiliations:** 1Dipartimento Scienze Della Vita e Dell’Ambiente, Università Politecnica Delle Marche, Via Brecce Bianche, 60131 Ancona, Italy; 2School of Food and Nutritional Sciences, University College Cork, T12 K8AF Cork, Ireland

**Keywords:** beer, non-conventional yeasts, lentil, functional beer

## Abstract

During the last few years, consumer demand has been increasingly oriented to fermented foods with functional properties. This work proposed to use selected non-conventional yeasts (NCY) *Lachancea*
*thermotolerans* and *Kazachstania*
*unispora* in pure and mixed fermentation to produce craft beer fortified with hydrolyzed red lentils (HRL). For this, fermentation trials using pils wort (PW) and pils wort added with HRL (PWL) were carried out. HRL in pils wort improved the fermentation kinetics both in mixed and pure fermentations without negatively affecting the main analytical characters. The addition of HRL determined a generalized increase in amino acids concentration in PW. *L. thermotolerans* and *K. unispora* affected the amino acid profile of beers (with and without adding HRL). The analysis of by-products and volatile compounds in PW trials revealed a significant increase of some higher alcohols with *L. thermotolerans* and ethyl butyrate with *K. unispora.* In PWL, the two NCY showed a different behavior: an increment of ethyl acetate (*K. unispora*) and β-phenyl ethanol (*L. thermotolerans*). Sensory analysis showed that the presence of HRL characterized all beers, increasing the perception of the fruity aroma in both pure and mixed fermentation.

## 1. Introduction

Nowadays, the growing interest in the craft beer market leads to a constant search for advanced processes, newly selected microorganisms, and raw materials leading to alternative products. Indeed, craft beers, often including “specialty beers”, have experienced exponential growth over the last two decades, primarily driven by premiumization and consumers’ willingness to seek new, intimate, and unique drinking experiences [1,2].

Although there are various options for obtaining different sensory profiles in craft beers (using special malts or adjuncts, hop varieties, water quality, etc.), the choice of yeast strains for wort fermentation and beer conditioning is crucial [3]. Indeed, most aroma-active compound production is strictly linked to the yeast strain that characterizes the beer in its style and final taste [4,5].

Although several yeast strains are commercially accessible, the availability of new starter strains remains an essential differentiating factor among craft beers produced in different microbreweries. Recently, as well as for enology, the brewing sector is growing the attention towards selected non-*Saccharomyces* yeasts for their possibility to confer unique aromas to the final product. This aspect has relevance considering craft beer as an unpasteurized, unfiltered, and re-fermented in-bottle beverage [6]. Specialty beers are products obtained following the classic style process with the addition of fruits, herbs and spices, various flavorings (e.g., liquorice, smoke, hot pepper), and alternative fermentable substrates (e.g., honey, maple syrup, molasses) [7]. Generally, specialty beers are all for beer styles which do not fit elsewhere. Low-calorie, low-alcohol or non-alcohol, novel-flavored, gluten-free, and functional beers are specialty beers of particular interest [8]. In addition, beer contains some health-promoting substances with positive impacts on the body, including minerals, vitamins, polyphenols, fiber, and relatively low levels of ethanol. Thus, beer can serve as a promising basis for developing a wide variety of functional beverages.

Functional beers are products obtained by adding beneficial health value, intended either as functional ingredients or functional fermenting yeasts [9]. An absolute novelty is represented by probiotic beer among the functional beers, obtained by incorporating probiotic microorganisms. Craft beer, an unpasteurized and unfiltered product, is potentially a vehicle for delivering probiotics. Chemical or biological acidification via *Lactobacilli* and bifidobacterial fermentation secure the microbial stability of the final beer. However, excessive levels of acetic acid could be produced if *Bifidobacteria* are incorporated.

Moreover, because viability is crucial for the efficacy of probiotics, attention must be paid to the sensitivity of probiotics to hop bitter acids, which can inhibit the survival of Gram-positive lactic acid bacteria [10,11]. Probiotics are not only bacteria; indeed, *Saccharomyces cerevisiae* var. *boulardii* is a probiotic yeast strain. Detailed information on the properties of probiotic yeast strains has been previously reported [12,13]. It will be of interest to explore these strains for specialty beer brewing. A novel unfiltered and unpasteurized probiotic beer could be produced by fermenting the wort with a probiotic strain of *S. cerevisiae*.

Studies have shown that foods and drinks with live probiotics are more effective in providing health effects than products containing inactive probiotics. Craft beer with live yeasts can be considered a new tool for beneficial health effects [14,15,16].

Another strategy to obtain functional beer could be to combine functional or probiotic yeasts with functional ingredients. Between them, a promising ingredient could be legumes such as chickpeas, lentils, and soy as an important source of protein for human nutrition. Legume products are essential in our daily diet to lead a healthy life [17,18]. From a nutritional point of view, legumes are of particular interest for the human diet as they are rich in fiber [19] and proteins: albumin and globulins are the dominant proteins present in legume seeds, with about 70% of legume proteins is produced by globulins [20,21]. Legumes also contain significant amounts of vitamins and other micronutrients. In this regard, the lentil (*Lens culinaris*) is a grain legume that represents an important protein source (25–30%). In Italy, lentils of Colfiorito and Castelluccio are an excellent product [22]. The goal of functional craft beer could be reached using wort enriched with hydrolyzed protein and fiber sources and by using probiotic yeasts [23].

In this scenario, the purpose of this study is to use non-conventional wild yeasts (NCY) to produce functional craft beer with reduced alcohol content. The selected NCY *Lachancea thermotolerans* and *Kazachstania unispora* were previously tested for their probiotic properties and subjected to safety assessment studies [24]. Fermentation trials were carried out using wort added with hydrolyzed red lentils (HRL) as a source of additional proteins and selected NCY in pure co-culture processes. The growth evolution, wort affinity, and viability during and after storage of selected NCY and the analytical, aromatic, and sensorial profile of functional beers were evaluated.

## 2. Materials and Methods

### 2.1. Yeast Strains

Three strains used in this study belong to the species *Lthermotolerans* (B13), *K.unispora* (M3-B3) and *S. cerevisiae* (2PV) coming from different un-anthropized environments and spontaneously processed foods. These strains were isolated, identified and characterized as probiotic and/or functional strains and tested in wort fermentation trials [23,24]. All of the yeast strains were maintained at 4 °C for short-term storage in YPD agar (yeast extract 10 g/L, peptone 20 g/L, dextrose 20 g/L, (agar 18 g/L) (Oxoid, Basingstoke, UK) and in YPD broth supplemented with 80% (*w*/*v*) glycerol at −80 °C, for long-term storage.

### 2.2. Fermentation Trials

*L. thermotolerans* and *K. unispora* were selected and used in pure and mixed fermentations with *S. cerevisiae*. In mixed fermentations *S. cerevisiae*/NYC yeast ratios were: 1:20 (*L. thermotolerans*), and 1:50 (*K. unispora*). The fermentation trials were carried out in 500-mL flasks containing 500 mL of wort at 20 ± 1 °C locked with a Müller hydraulic valve. The flasks were inoculated with 72 h pre-cultures grown in 10% malt extract at 20 ± 1 °C. The fermentation kinetics were monitored by measuring the weight loss of the flasks due to the CO_2_ evolved until the end of the fermentation (i.e., constant weight for 3 consecutive days). The fermentations were carried out in triplicate trials under static conditions.

### 2.3. Pils Wort and Lentil Wort Preparation

The trials were conducted in two worts: pils wort (PW) and pils wort added with HRL (PWL). PW comes from a batch of 1500 L of malted barley wort to produce Pilsner beer with the following main analytical characters: pH 5.4, specific gravity 12.2 °Plato, and 20 IBU. Two set of fermentation trials were conducted to evaluate the fermentation potential of yeast strains. First, PWL was prepared using pils wort and HRL. HRL was prepared using a mixture of lentil flour (70%) and water with the addition of α-amylase Hitempase STXL Kerry Group (Tralee, Ireland); and proteolytic enzyme Bioprotease P1 from Kerry Group (Tralee, Ireland). The washing procedures were conducted following the protocol reported by Canonico et al. [23] (1 h, from 45 °C to 75 °C) After that the substrate was boiled and centrifuged, obtaining the resulting wort that was added at 20% to PW.

### 2.4. By-Products and Volatile Compounds

(Glucose, sucrose, maltose were determined using specific enzymatic kits (kit k-masug) Megazyme, Wicklow, Ireland), while the protein content in final beers was measured using Lowry method. [25]. Direct injection of final beers prepared following Canonico et al. [26] procedure into a gas chromatography system (GC-2014; Shimadzu, Kjoto, Japan) was used to quantify acetaldehyde ethyl acetate, n-propanol, isobutanol, amyl and isoamyl alcohols. The main volatile compounds were determined by solid-phase microextraction (HS-SPME) method using a fiber Divinylbenzene/Carboxen/Polydimethylsiloxane (DVB/CAR/PDMS) (Sigma-Aldrich, St. Louis, MO, USA). The compounds were desorbed by inserting the fiber into gas chromatograph GC (GC-2014; Shimadzu, Kjoto, Japan) identified and quantified by comparisons with external calibration curves for each compound [26].

### 2.5. Sensory Analysis

The secondary fermentation was carried out in 330 mL bottles with the addition of 5.5 g/L of sucrose at 18–20 °C for 7–10 days. After bottle re-fermentation, the beers were stored at 4 °C and underwent sensory analysis using a scale from 1 to 10 [27] based on a list of descriptors related to both the aromatic notes (e.g., floral, fruity, toasty) and the main structural features (e.g., sweet, acidity, flavor, astringency, bitterness, olfactory persistence). A group of 10 trained testers carried this out. The data were elaborated with statistical analyses to obtained information about the contribution on each descriptor on the organoleptic quality of beer. Informed consent was obtained from all subjects involved in the study of sensory analysis.

### 2.6. Yeast Vitality Assay after 3 Months of Bottling

The vitality of the strains after 3 months of bottling was carried out using viable cell counts on WL Nutrient Agar (Oxoid, Hampshire, UK) and Lysine Agar (Oxoid, Hampshire, UK) for the differentiation of NCY yeast from *S. cerevisiae* strain.

### 2.7. Nutritional Values Amino Acid Composition of Final Beers

Determination of moisture and dry matter in food for human use by gravimetry was determined by Method/Document acronym: ISTISAN 1996/34 Met B; Method/Document title: Calculation of Carbohydrates and Energy Value in food for human use was determined by Method/Document acronym: MP 0297 rev 6 2018; Method/Document acronym: Determination of amino acids in food, fertilizers and soil improvers by ion chromatography was determined by Method/Document title: MP 1471 rev 6 202; Method/Document title: Determination of ash/crude ash in food for human and zootechnical use by gravimetry was determined by Method/Document acronym: MP 2271 rev 0 2018; Method/Document title: Determination of total fat substances in food for human use by gravimetry (method with acid hydrolysis) was determined by Method/Document acronym: ISTISAN 1996/34 Met A.

### 2.8. Statistical Analysis

Analysis of variance (ANOVA) was applied to the experimental data for the main analytical character of the beers. The significant differences were determined using Duncan tests, and the results were considered significant if the associated *p* values were <0.05. Principal component analysis (PCA) was applied to discriminate between the means of the contents of volatile compounds. The statistical software package JMP 11 ^®^ was used for statistical analysis.

## 3. Results

### 3.1. Fermentation Kinetics

Figure 1 shows the fermentation kinetics of *S. cerevisiae*, *K. unispora,* and *L. thermotolerans* strains in pure and in mixed fermentation (*S. cerevisiae*/*K. unispora* and *S. cerevisiae*/*L. thermotolerans*), in both PW (Figure 1a) and PWL (Figure 1b).

*K. unispora* and *L. thermotolerans* in pure fermentation showed slower fermentation kinetics than *S. cerevisiae* pure culture, both in their respective mixed fermentations and in both worts tested together. As expected, *S. cerevisiae* pure culture showed the highest fermentation kinetics compared to the other fermentation trials, with the maximum loss of CO_2_ within the first 3–4 days of fermentation in both PW and PWL worts. However, mixed fermentation (in both worts) and the presence of lentil (PWL)-enhanced fermentation kinetics of *K. unispora* and *L. thermotolerans* pure fermentations.

### 3.2. Main Analytical Characteristics

The data of the main analytical characters of the beers obtained at the end of the primary fermentation on PW and PWL are reported Table 1.

Glucose and sucrose were completely consumed in all fermentation trials, while only *S. cerevisiae* trials did not show residual maltose in both PW and PWL. Pure and mixed NCY fermentation trials showed residual maltose. This residue was similar in all PW fermentation trials (12–13 g/L), while *K. unispora* pure fermentation showed higher residual maltose (30 g/L), indicating a lower fermentation activity. In PWL a general enhancement of maltose consumption in NCY fermentation trials was shown. The ethanol content was generally lower with NCY in pure and mixed fermentations, even if this reduction is significant only in a few cases. No significant difference between the final beer was shown regarding the other analytical character.

Regarding the amino acid profile (Table 2), the final beer on PW exhibited a significant increase in aspartic acid and phenylalanine content with *K. unispora,* in both pure and mixed fermentations.

Still, in PW *L. thermotolerans* pure fermentation exhibited a significant increase in glutamic acid, lysine, and asparagine content. The other amino acids did not show significant differences among the trials tested. As expected, the final beers with lentil (PWL) exhibited a higher content in all amino acids tested compared to PW. Among the trials carried out in PWL, *L. thermotolerans* pure and mixed fermentations exhibited a general significant reduction in amino acid content (except for phenylalanine, methionine, and ornithine), while *K. unispora* (pure and mixed fermentations) exhibited a similar amino acid content with the *S. cerevisiae* pure culture.

To assess the influence of *L. thermotolerans* and *K. unispora* on both wort trials, the data were elaborated by principal component analysis (PCA) (Figure 2).

The trials carried out on PW (Figure 2a), *K. unispora* pure culture and mixed fermentations grouped in the lower right quadrant showed that *K. unispora* characterized the final amino acid content in mixed culture. Different behavior was exhibited by *L. thermotolerans* that in pure culture was in the right upper quadrant, in mixed (*L. thermotolerans*/*S. cerevisiae*) in the lower left quadrant, while *S. cerevisiae* pure culture was in an intermediate position highlighting a different effect of this yeast on the final amino acid content of the beer. The distribution of the trials carried out on PWL (Figure 2b) showed a more homogeneous distribution among pure and mixed fermentations. Indeed, *K. unispora* pure and mixed fermentations were in the right upper quadrant, *L. thermotolerans* pure and mixed trials on the line that separates the two left quadrants and *S. cerevisiae* in the lower right quadrant. These data indicated that *K. unispora* and *L. thermotolerans* affected the beer’s final amino acid composition compared with *S. cerevisiae*.

### 3.3. By-Products and Volatile Profiles

The data for the main by-products and the volatile compounds in PW and PWL are reported in Table 3. The results of PW trials indicated that *L. thermotolerans* and *K. unispora* in mixed fermentation increased some aroma compounds compared to pure culture. *L. thermotolerans*/*S. cerevisiae* significantly increased the n-propanol, isobutanol, and amylic alcohol, while *K. unispora*/*S. cerevisiae* significantly increased ethyl butyrate in comparison with the other trials.

A different trend was shown in PWL, where the *K. unispora* pure culture increased ethyl acetate, while isobutanol and isoamyl acetate were comparable to that exhibited by *K. unispora*/*S. cerevisiae,* and *L. thermotolerans*/*S. cerevisiae,* respectively. Moreover, *L. thermotolerans*/*S. cerevisiae* led to a significant increase in β-phenyl ethanol compared to the other trials. *L. thermotolerans* showed a significant enhancement of acetaldehyde content, while *S. cerevisiae* was characterized by ethyl butyrate, ethyl acetate, and amylic and isoamylic alcohol.

To assess the overall effects of yeast strains, modalities of inoculum, and different worts used, the data of the by-products, volatile compounds were analyzed by PCA (Figure 3).

The graphic representation of PCA of the fermentation products responsible for beer’s aroma showed a clear separation of beers obtained by two different worts. The PW trials were in the upper left quadrant while PWL trials were in the lower right quadrant, showing a less homogeneous distribution. Indeed, the fermentation with *S. cerevisiae* pure culture was in the upper right quadrant. The two non-*Saccharomyces* in pure and mixed fermentations were in the opposite quadrant (lower right quadrant). Furthermore, PCA analysis showed a separation between pure and mixed culture, highlighting an effect of non-*Saccharomyces* strains and their inoculation modality on the volatile composition of beer. In PWL NCY (pure and mixed fermentations) strongly characterized the aromatic profile resulting in more distance in the graphical distribution by *S. cerevisiae*.

### 3.4. Vitality Assay after 3 Months

The vitality assay of all strain tested (data not shown) showed a good vitality after 3 months of bottling, exhibiting a viable cell count c.a ≥ 6.5 log CFU/mL in both substrates and compared with the initial inoculum (c.a. 6 log CFU/mL). This is a promising result since the vitality after storage is very closed to that fixed for probiotic bacteria claim.

### 3.5. Sensorial Analysis

The beers obtained by pure and mixed fermentations on PW and PWL underwent sensory analysis, and the results are reported in Figure 4.

All beers analyzed showed differences for their main aromatic notes. The fermentation carried out with *K. unispora* pure fermentation on PW (Figure 4a) showed a significant difference in the perception of alcoholic solvent, malt note, sweetness and persistence. This last feature was also significantly emphasized in S. cerevisiae pure culture, which also exhibited a perception of the descriptor “other sulfide”. *L. thermotolerans* showed malty and fruity/citric notes. Overall, all the beers are characterized by distinctive and characterizing notes.

Regarding beer brewed with PWL (Figure 4b), in *L. thermotolerans* pure cultures the presence of lentils increased the perception of fruity esters, fruity/citric, astringency, and acidity. Additionally, *K. unispora*/*S. cerevisiae* emphasized the fruity/citric notes. Moreover, all fermentations exhibited persistence, and in all single fermentations, an increase in DMS was observed

## 4. Discussion

In recent years, the attention of researchers has been focused not only on the exploitation of new raw materials to obtain a distinctive beer but also on the potential health benefits for consumers [28]. Indeed, beer is rich in bioactive compounds coming from traditional ingredients (barley, hop, and yeast) but also from special ingredients such as spices, cereal, herbs, fruits, and legumes, which can affect the nutritional composition of the final product [23,29]. Although the health benefits of fermented beverages are well established in the scientific field, this is not equally perceived in public opinion. Nevertheless, the positive impact of yeast on consumer health can be related to several aspects: providing a source of probiotic microbes; providing prebiotic metabolites through the secondary metabolism of compounds derived from the grain, hops, or other ingredients; and the production of antimicrobial compounds [30,31].

In this work, selected yeasts with functional aptitude [23,24] were evaluated in pure and mixed culture on PW and PWL added with HRL. The goal of craft beer with functional properties and strengthen the sensory profile was obtained through a double strategy: on the one hand to provide for the increase in vegetable protein and on the other for the use of probiotic fermenting yeasts.

As expected, the addition 20% of HRL determined an enhancement of total protein concentration of craft beer supporting the recent trend of consumers demanding protein-enriched foods. Moreover, the hydrolysis process enriches the wort with essential amino acids that contribute to improving the nutritional profile of the final product. On the other hand, this addition increased the fermentation performance of yeasts both in pure and mix cultures, highlighting the possible use of this substrate in the brewing process to improve the overall fermentation process confirming previous preliminary findings [24]. Furthermore, changes in the concentrations of amino acids in the fortified wort, influencing the nitrogen metabolism, led to a greater availability of amino acids determining an enhancement of the fermentation activity, specifically of *L. thermotolerans* and *K. unispora* strains. In this regard, Krasnikova et al. [32] found an enhancement of enzymatic activity in *S. cerevisiae* due to the supplementation of wort with nitrogen sources from lentils.

The addition of HRL showed a limited increase in maltose concentration with a consequent increase in ethanol content. Over the last decade, plant-based beverages have gained popularity amongst aware consumers seeking alternative and environmentally sustainable options to traditional drinks. Recently, Nawaz et al. [33] reviewed the involvement of fermented yeasts in alternative substrates in the emerging segment of the functional legume-based beverages. They highlighted the effective opportunities to broaden and diversify new products, characterized by legume addition, which may offer better nutrition content and distinctive taste.

In brewing, the composition of wort is an essential part of beer flavor. In this regard, although wort is complex and not thoroughly characterized, the content of the amino acid indubitably affected the production of some minor metabolic products of fermentation which contribute to the flavor of the beer. The addition of a protein source such as HRL determined an effective improvement of aromatic compounds such as higher alcohols, esters, carbonyl and sulfur compounds. Except for the preliminary screening work [23], no other published work has investigated the use of hydrolyzed lentils to produce functional craft beer. The distinctive footprint found in pure and mixed fermentation with PWL could be partially related to the yeast catabolism of increased availability of amino acids. During wort fermentation, amino acid utilization by yeast is closely linked to flavour profile [34] such as higher alcohols and related esters. Thus, an improved understanding of amino acid uptake and assimilation is essential to generate defined amounts of metabolites to regulate specific sensory perception in fermented beverages.

Despite the high levels of nitrogen and/or free amino acids coming from HRL, in PWL trials the final flavors were characterized by specific amino acids, such as glutamate, aspartate, and asparagine. In line these results, Black and co-workers [35] highlighted the positive effect of fava bean that changed the proportion of unfavorable amino acids in brewing. On the other hand, the use of NCY in pure and mixed cultures positively influenced the aromatic and sensorial profile, particularly in fortified beers. The interest in NCY isolated from various food and the environmental source is another strategy to bring innovation in the brewing sector. Brewers widely seek the ability to improve the analytical and aromatic profiles during alcoholic fermentation [3,36,37,38,39,40]. In this work we evaluated the dual role of yeasts: metabolizing the amino acids characterizing the vegetal notes (unwanted) and using them as precursors of aromatic compounds.

Differently from *L. thermotolerans*, yeast species extensively investigated in the production of beer [36,38,39,40,41], *K. unispora* was only recently proposed [23]. This yeast is usually encountered at low frequency in natural environments; the genus *Kazachstania* was first discovered in 1971 with *Kazachstania viticola* isolated from fermenting grapes. Subsequently, other species such as *K. unispora* (formerly *Saccharomyces unisporus*) were found on grape must in different countries [42], although its metabolic footprint remains widely unknown. Mixed fermentations with NCY lead final products with increased persistence and astringency and a general exalted fruity note [23]. Moreover, all strains tested showed a good vitality in both substrates after bottle re-fermentation and storage. The viability found here was close to 7 Log/mL(claim for probiotic bacteria) indicating the potential functional trait of NCY *K. unispora* and *L. thermotolerans* here tested. Further investigations are necessary to confirm the positive role of these non-conventional yeasts at the human-gut-level, combining consumer acceptability.

## 5. Conclusions

This study showed that the combined use of selected NCY and the addition of HRL in pils wort could be a suitable strategy to manage a specialty craft beer with enhanced functional properties and a promising sensory profile.

The different amino acid metabolism of NCY tested led the aromatic profile of the fortified beers with distinctive and positive sensory notes.

The promising results indicate a possible market exploitation of such innovative fermented beverages. Indeed, the conjunction of a legume fortified beer and the presence of functional yeasts could represent a great opportunity to put an innovative product on the market.

## Figures and Tables

**Figure 1 foods-11-02787-f001:**
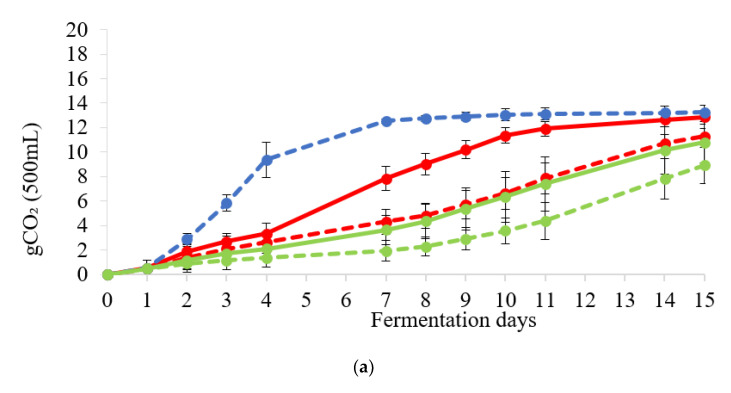
Fermentation kinetics of the pure and mixed fermentations in (**a**) PW and (**b**) PWL. Pure culture of *S. cerevisiae* (
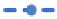
); *L. thermotolerans* (
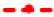
), and *K. unispora* (
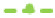
), and mixed fermentation of *S. cerevisiae*/*L. thermotolerans* (
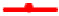
) and *S. cerevisiae*/*K. unispora* (
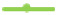
).

**Figure 2 foods-11-02787-f002:**
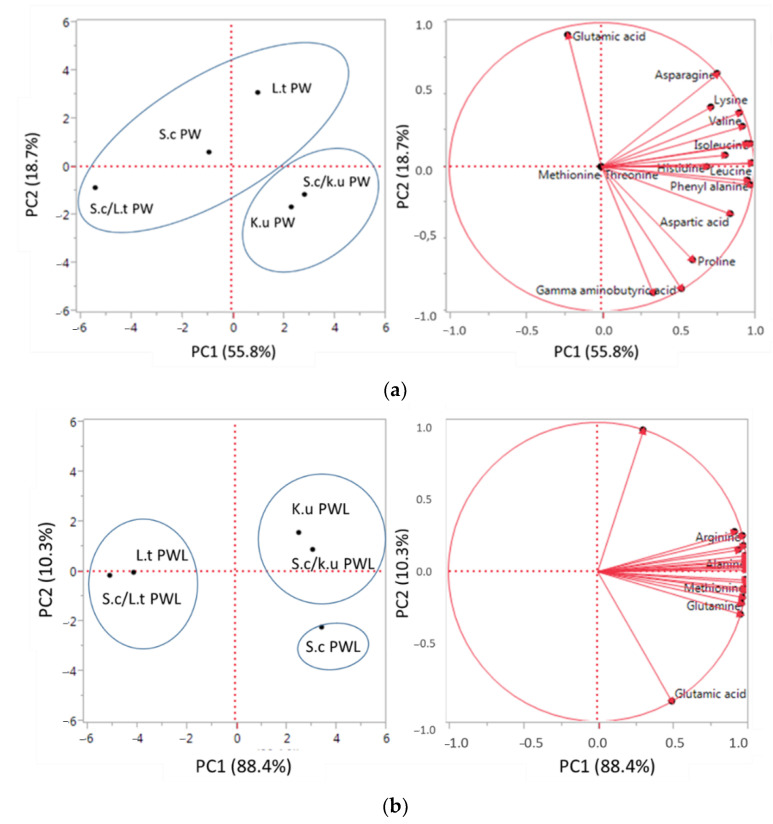
Principal component analysis of amino acid content of final beers. (**a**) Pils wort (PW): the variance explained by principal component analysis (PCA) analysis is PC 1 55.8% X-axis and PC 2 18.7% Y-axis. (**b**) Pils wort + lentil (PWL): the variance explained by principal component analysis (PCA) analysis is PC 1 88.4% X-axis and PC 2 10.3% Y-axis.

**Figure 3 foods-11-02787-f003:**
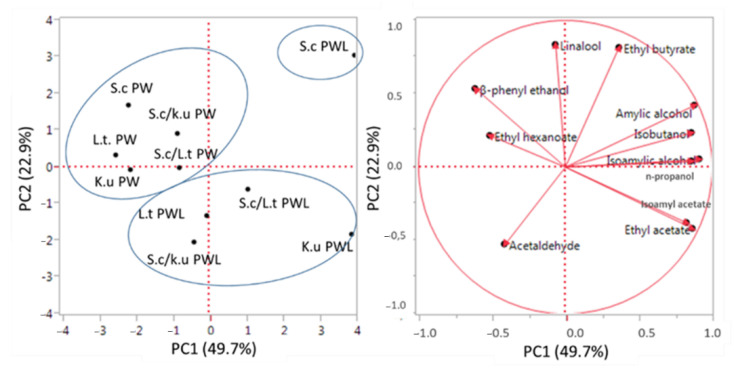
Principal component analysis for the main by-products and volatile compounds of craft beer obtained by different yeast strains in PW and PWL. The variance explained by principal component analysis (PCA) analysis is PC 1 49.7% X-axis and PC 2 22.9% Y-axis.

**Figure 4 foods-11-02787-f004:**
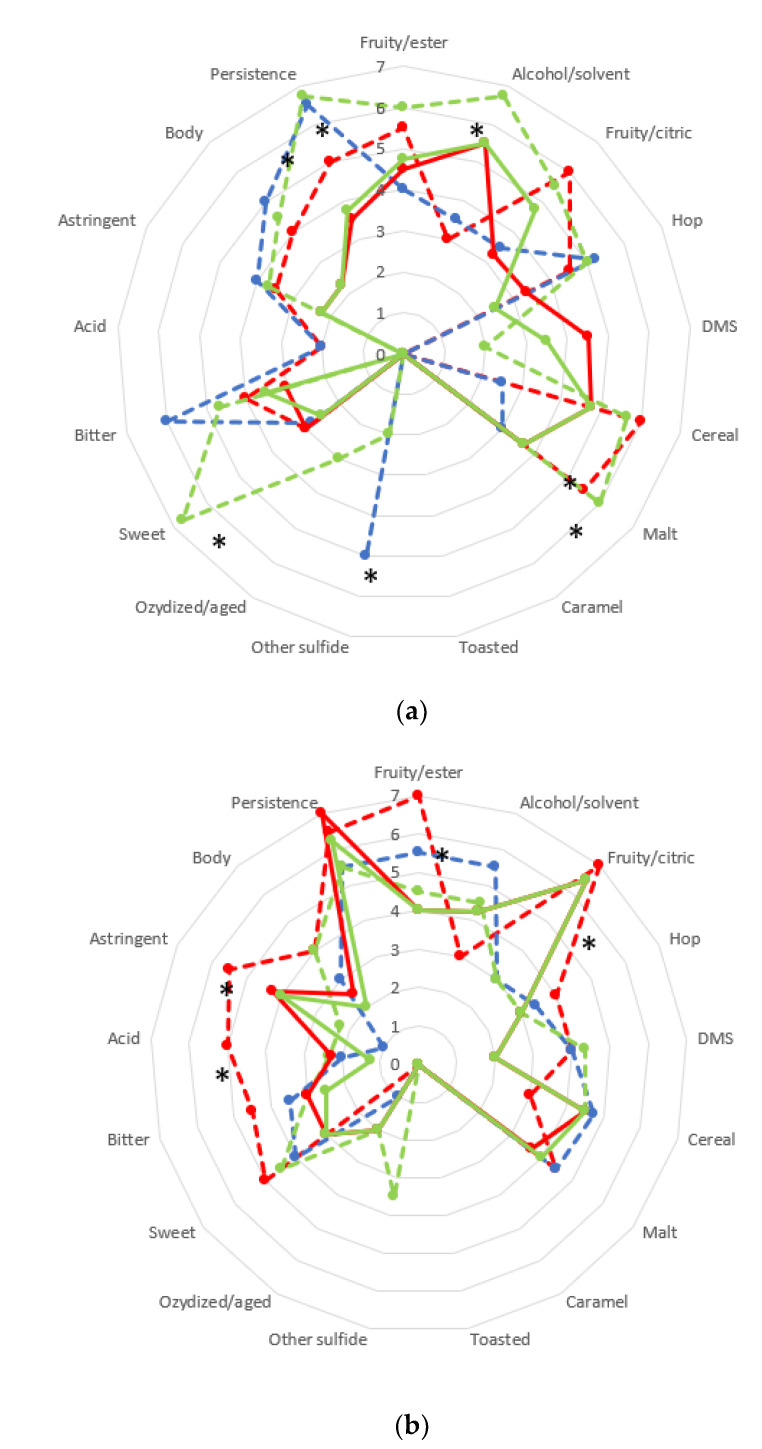
Sensory analysis of the beer produced by pure and mixed fermentation on pils wort (**a**) and pils wort added lentil (**b**). Pure culture of *S. cerevisiae* (
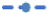
), *L. thermotolerans* (
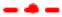
) and *K. unispora* (
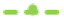
), and mixed fermentation of *S. cerevisiae*/*L. thermotolerans* (
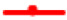
) and *S. cerevisiae*/*K. unispora* (
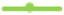
). * significantly different (Fisher ANOVA; ***p***-value 0.05). DMS, Dimethyl sulphide. Score 0: absence of the descriptor analyzed.

**Table 1 foods-11-02787-t001:** The main analytical characters of final beer on PW (pils wort) and PWL (pils wort added with 20% HRL). Data are the means ± standard deviations. Data with different superscript letters (^a, b, c^) within each row between the fermentation trials tested on same wort (Duncan tests; *p* < 0.05). The initial composition of the sugars in pils wort were: Glucose 11.92 g/L; Sucrose 24.2 g/L; Maltose 74.73 g/L. Protein content: 12.02 g/L. The initial composition of the sugars in pils wort added with lentil were: Glucose 6.74 g/L; Sucrose 37.72 g/L; Maltose 84.70 g/L. Protein content: 26.39 g/L. %). ND: not detected. LoQ: limit of quantification (1 g/L).

			*PW*					*PWL*		
The Main Fermentation Parameters	*S. cerevisiae*	*L. thermotolerans*	*K. unispora*	*S. cerevisiae/L. thermotolerans*	*S. cerevisiae/K. unispora*	*S. cerevisiae*	*L. thermotolerans*	*K. unispora*	*S. cerevisiae/* *L. thermotolerans*	*S. cerevisiae/* *K. unispora*
**Residual Glucose** **g/L**	ND	ND	ND	ND	ND	ND	ND	ND	ND	ND
**Residual Sucrose** **g/L**	ND	ND	ND	ND	ND	ND	ND	ND	ND	ND
**Residual Maltose** **g/L**	ND	13.52 ± 1.50 ^b^	30.14 ± 0.65 ^a^	12.91 ± 0.15 ^b^	11.95 ± 1.61 ^b^	0.11 ± 1.10 ^c^	4.77 ± 0.1 ^b^	4.44 ± 2.164 ^b^	5.97 ± 0.704 ^b^	11.95 ± 1.61 ^a^
**Protein** **g/L**	12.53 ± 3.87 ^a^	33.29 ± 3.71 ^a^	25.51 ± 16.45 ^a^	18.49 ± 11.32 ^a^	17.43 ± 13.18 ^a^	26.27 ± 12.99 ^a^	30.12 ± 3.89 ^a^	25.51 ± 11.68 ^a^	22.05 ± 8.43 ^a^	17.43 ± 13.18 ^a^
**Ethanol** **%*v*/*v***	3.38 ± 0.14 ^a^	3.04 ± 0.1 ^a, b^	3.03 ± 0.12 ^a, b^	2.99 ± 0.10 ^b^	3.07 ± 0.18 ^a, b^	3.76 ± 0.17 ^a^	3.3 ± 0.17 ^a, b^	3.78 ± 0.10 ^a^	3.44 ± 0.06 ^a, b^	3.07 ± 0.18 ^b^
**Moisture** **g/100 g**	95.90 ± 0.38 ^a^	95.73 ± 0.38 ^a^	95.85 ± 0.38 ^a^	96.18 ± 0.38 ^a^	95.89 ± 0.38 ^a^	94.19 ± 0.38 ^a, b^	93.82 ± 0.38 ^b^	94.34 ± 0.38 ^a, b^	93.95 ± 0.38 ^a, b^	95.89 ± 0.38 ^a^
**Fatty** **g/100 g**	0.050 ± 0.033 ^a^	<LoQ ^a^	<LoQ ^a^	<LoQ ^a^	<LoQ ^a^	<LoQ ^a^	<LoQ ^a^	<LoQ ^a^	<LoQ ^a^	<LoQ ^a^
**Ashes** **g/100 g**	0.15 ± 0.03 ^a^	0.17 ± 0.03 ^a^	0.19 ± 0.04 ^a^	0.19 ± 0.04 ^a^	0.20 ± 0.04 ^a^	03.0 ± 00.4 ^a^	0.29 ± 0.04 ^a^	0.34 ± 0.04 ^a^	0.30 ± 0.04 ^a^	0.20 ± 0.04 ^a^
**Carbohydrates** **g/100 g**	3.43 ± 0.39 ^a^	3.63 ± 0.39 ^a^	3.50 ± 0.39 ^a^	3.14 ± 0.39 ^a^	3.47 ± 0.39 ^a^	4.30 ± 0.39 ^a^	4.74 ± 03.9 ^a^	4.15 ± 0.39 ^a^	4.63 ± 0.39 ^a^	3.47 ± 0.39 ^a^
**Energy value** **kcal/100 g**	16 ± 2 ^a^	16 ± 2 ^a^	16 ± 2 ^a^	15 ± 2 ^a^	16 ± 2 ^a^	22 ± 2 ^a, b^	24 ± 2 ^a^	21 ± 2 ^a, b^	23 ± 2 ^a, b^	16 ± 2 ^b^
**Dry substance** **g/100 g**	4.10 ± 0.38 ^b^	4.27 ± 0.38 ^b^	6.7 ± 0.38 ^a^	3.82 ± 0.38 ^b^	4.11 ± 0.38 ^b^	5.81 ± 0.38 ^a^	6.18 ± 0.38 ^a^	5.66 ± 0.38 ^a^	6.05 ± 0.38 ^a^	4.11 ± 0.38 ^b^

**Table 2 foods-11-02787-t002:** Amino acid composition of final beers produced in pure and mixed fermentations on PW (pils wort) and (PWL) pils wort added with lentil. Data are means ± standard deviations from three independent experiments. Data with different superscript letters (^a, b, c^) within each row between the fermentation trials tested on the same wort (Duncan tests (0.05%). LoQ: limit of quantification (1 g/L).

			*PW (Pils Wort)*					*PWL (Pils + Lentil Wort)*		
Amino Acid Composition (mg/L)	*S. cerevisiae*	*L. thermotolerans*	*K. unispora*	*S. cerevisie/L. thermotolerans*	*S. cerevisiae/K. unispora*	*S. cerevisiae*	*L. thermotolerans*	*K. unispora*	*S. cerevisie/L. thermotolerans*	*S. cerevisiae/K. unispora*
Aspartic acid	45 ± 8 ^a, b^	31 ± 7 ^a, b^	49 ± 8 ^a^	18 ± 7 ^b^	54 ± 9 ^a^	257 ± 26 ^a^	110 ± 13 ^b^	226 ± 23 ^a^	105 ± 12 ^b^	224 ± 23 ^a^
Glutamic acid	53 ± 8 ^a, b^	56 ± 9 ^a^	26 ± 8 ^b^	36 ± 8 ^a, b^	28 ± 7 ^a, b^	495 ± 48 ^a^	246 ± 25 ^b^	220 ± 23 ^b^	234 ± 24 ^b^	277 ± 28 ^b^
Alanine	80 ± 11 ^a^	81 ± 10 ^a^	81 ± 10 ^a^	66 ± 9 ^a^	84 ± 11 ^a^	318 ± 31 ^a^	218 ± 22 ^a, b^	309 ± 31 ^a, b^	209 ± 21 ^b^	313 ± 31 ^a^
Arginine	72 ± 10 ^a^	69 ± 10 ^a^	79 ± 11 ^a^	48 ± 8 ^a^	80 ± 10 ^a^	277 ± 27 ^a, b^	191 ± 20 ^b^	303 ± 30 ^a^	180 ± 19 ^b^	303 ± 30 ^a^
Asparagine	18 ± 7 ^a, b^	31 ± 7 ^a^	16 ± 7 ^a, b^	<LoQ ^b^	21 ± 7 ^a, b^	135 ± 15 ^a, b^	105 ± 13 ^a, b^	134 ± 15 ^a, b^	84 ± 11 ^b^	151 ± 16 ^a^
Proline	319 ± 32 ^a^	326 ± 3 ^a^	349 ± 35 ^a^	327 ± 32 ^a^	341 ± 34 ^a^	369 ± 37 ^a^	330 ± 33 ^a^	392 ± 39 ^a^	323 ± 32 ^a^	368 ± 36 ^a^
Phenyl alanine	32 ± 8 ^a, b^	44 ± 8 ^a, b^	56 ± 9 ^a^	17 ± 7 ^b^	54 ± 9 ^a^	219 ± 22 ^a^	154 ± 16 ^a, b^	215 ± 22 ^a^	142 ± 16 ^b^	220 ± 22 ^a^
Glycine	29 ± 7 ^a^	26 ± 7 ^a^	30 ± 7 ^a^	28 ± 7 ^a^	30 ± 7 ^a^	138 ± 15 ^a^	71 ± 10 ^b^	118 ± 14 ^a^	72 ± 10 ^b^	116 ± 13 ^a, b^
Glutamine	15 ± 7 ^a^	10 ± 7 ^a^	12 ± 7 ^a^	<LoQ ^a^	14 ± 7 ^a^	65 ± 9 ^a^	13 ± 7 ^b^	51 ± 8 ^a^	16 ± 7 ^b^	52 ± 8 ^a^
Isoleucine	14 ± 7 ^a^	19 ± 7 ^a^	20 ± 7 ^a^	<LoQ ^a^	20 ± 7 ^a^	144 ± 16 ^a^	89 ± 11 ^b^	140 ± 15 ^a^	75 ± 10 ^b^	151 ± 16 ^a^
Histidine	12 ± 7 ^a^	30 ± 8 ^a^	26 ± 7 ^a^	18 ± 7 ^a^	35 ± 8 ^a^	78 ± 10 ^a^	44 ± 8 ^b^	63 ± 9 ^a, b^	43 ± 8 ^b^	68 ± 9 ^a, b^
Leucine	30 ± 7 ^a^	38 ± 8 ^a^	42 ± 8 ^a^	17 ± 7 ^b^	42 ± 8 ^a^	260 ± 26 ^a^	166 ± 18 ^b^	251 ± 25 ^a^	151 ± 16 ^b^	264 ± 27 ^a^
Lysine	<LoQ ^b^	20 ± 7 ^a^	12 ± 7 ^a, b^	<LoQ ^b^	12 ± 7 ^a, b^	206 ± 21 ^a^	112 ± 13 ^b^	208 ± 21 ^a^	101 ± 12 ^b^	218 ± 22 ^a^
Methionine	<LoQ	<LoQ	<LoQ	<LoQ	<LoQ	59 ± 9 ^a^	39 ± 8 ^a^	54 ± 9 ^a^	34 ± 8 ^a^	58 ± 9 ^a^
Ornithine	<LoQ	<LoQ	<LoQ	<LoQ	<LoQ	20 ± 7 ^a^	14 ± 7 ^a^	21 ± 7 ^a^	14 ± 7 ^a^	21 ± 7 ^a^
Serine	14 ± 7 ^a^	18 ± 7 ^a^	14 ± 7 ^a^	<LoQ	17 ± 7 ^a^	192 ± 20 ^a^	103 ± 12 ^b^	183 ± 19 ^a^	87 ± 11 ^b^	193 ± 20 ^a^
Tyrosine	49 ± 8 ^a^	61 ± 9 ^a^	66 ± 10 ^a^	36 ± 8 ^a^	64 ± 9 ^a^	209 ± 21 ^a^	141 ± 16 ^b^	193 ± 20 ^a, b^	135 ± 15 ^b^	192 ± 20 ^a, b^
Threonine	<LoQ	<LoQ	<LoQ	<LoQ	<LoQ	98 ± 12 ^a^	48 ± 8 ^b^	95 ± 11 ^a^	37 ± 8 ^b^	106 ± 12 ^a^
Valine	39 ± 8 ^a^	56 ± 9 ^a^	52 ± 8 ^a^	28 ± 7 ^a^	52 ± 9 ^a^	304 ± 30 ^a^	200 ± 20 ^b, c^	280 ± 28 ^a, b^	183 ± 19 ^c^	292 ± 29 ^a, b^
Gamma aminobutyric acid	71 ± 10 ^a^	64 ± 9 ^a^	82 ± 10 ^a^	70 ± 10 ^a^	82 ± 10 ^a^	136 ± 15 ^b^	104 ± 12 ^b^	283 ± 28 ^a^	104 ± 12 ^b^	242 ± 24 ^a^
Total free amino acid	892 ± 45 ^a^	980 ± 46 ^a^	1012 ± 48 ^a^	709 ± 41 ^b^	1030 ± 48 ^a^	3979 ± 104 ^a^	2498 ± 72 ^c^	3739 ± 97 ^a, b^	2 329 ± 68 ^c^	3829 ± 97 ^b^

**Table 3 foods-11-02787-t003:** The main by-products and volatile compounds (mg/L) of pure and mixed fermentation trials carried out in PW (pils wort) and PWL (pils wort with HRL). Data are means ± standard deviations from three independent experiments. Data with different superscript letters (^a, b, c, d^) within each row between the fermentation trials tested on same wort (Duncan tests; *p* < 0.05). * OTVs (odor threshold value; mg/L).

			*PW*					*PWL*		
The Main By-Products (OTVs *)	*S. cerevisiae*	*L. thermotolerans*	*K. unispora*	*S. cerevisie/L. thermotolerans*	*S. cerevisiae/K. unispora*	*S. cerevisiae*	*L. thermotolerans*	*K. unispora*	*S. cerevisie/L. thermotolerans*	*S. cerevisiae/K. unispora*
**Ethyl butyrate** (0.14–0.37)	0.117 ± 0.025 ^b^	0.067 ± 0.021 ^c^	0.053 ± 0.007 ^c^	0.083 ± 0.068 ^b, c^	0.268 ± 0.016 ^a^	0.335 ± 0.029 ^a^	0.016 ± 0.009 ^c^	0.055 ± 0.028 ^b^	0.087 ± 0.037 ^b^	0.060 ± 0.006 ^b^
**Ethyl acetate** (7.5–31)	4.04 ± 0.52 ^c^	7.08 ± 0.55 ^a^	6.01 ± 0.89 ^b, c^	7.74 ± 0.75 ^a^	6.26 ± 0.47 ^a, b^	15.81 ± 1.04 ^a, b^	15.64 ± 1.08 ^a, b^	20.54 ± 0.88 ^a^	15.29 ± 3.43 ^a, b^	11.73 ± 2.38 ^b^
**Linalool** (0.0006–0.001)	0.047 ± 0.029 ^a^	0.046 ± 0.022 ^a^	0.037 ± 0.004 ^a^	0.028 ± 0.020 ^a^	0.031 ± 0.012 ^a^	0.055 ± 0.033 ^a^	0.031 ± 0.008 ^a, b^	0.024 ± 0.002 ^b^	0.032 ± 0.004 ^a, b^	0.028 ± 0.00 ^a, b^
**Ethyl hexanoate** (0.17–0.20)	0.027 ± 0.009 ^a^	0.025 ± 0.005 ^a^	0.028 ± 0.004 ^a^	0.031 ± 0.00 ^a^	0.020 ± 0.007 ^a^	0.025 ± 0.004 ^a^	0.023 ± 0.001 ^a^	0.014 ± 0.007 ^a^	0.019 ± 0.004 ^a^	0.032 ± 0.010 ^a^
**Isoamyl acetate** (0.30–0.72)	0.40 ± 0.05 ^a^	0.11 ± 0.18 ^b^	0.26 ± 0.22 ^a, b^	0.32 ± 0.03 ^a, b^	0.40 ± 0.01 ^a^	0.820 ± 0.143 ^b^	0.527 ± 0.021 ^c^	1.514 ± 0.349 ^a^	1.070 ± 0.063 ^a^	0.851 ± 0.065 ^b^
**n-propanol** (0.8–5.0)	18.09 ± 1.12 ^b^	16.17 ± 1.01 ^b^	17.01 ± 1.13 ^b^	22.85 ± 1.69 ^a^	18.35 ± 0.96 ^b^	25.85 ± 2.18 ^a^	18.28 ± 2.43 ^c^	24.64 ± 1.70 ^a, b^	20.92 ± 2.59 ^b, c^	23.37 ± 3.20 ^a, b, c^
**Isobutanol** (3.2–14.5)	8.838 ± 0.542 ^a, b^	7.425 ± 0.510 ^c^	9.937 ± 0.628 ^b, c^	12.70 ± 2.36 ^a^	11.19 ± 0.40 ^a, b^	24.51 ± 3.69 ^a^	13.56 ± 1.64 ^b^	20.88 ± 2.48 ^a^	12.52 ± 2.33 ^b^	23.37 ± 3.20 ^a^
**Amylic alcohol** (0.32–15.0)	6.722 ± 0.572 ^a, b^	5.336 ± 0.104 ^c^	6.404 ± 1.048 ^b, c^	7.756 ± 0.491 ^a^	7.689 ± 0.239 ^a^	15.67 ± 2.01 ^a^	6.727 ± 0.098 ^c^	10.65 ± 0.92 ^b^	7.822 ± 1.657 ^c^	8.140 ± 1.755 ^b, c^
**Isoamylic alcohol** (0.77–16.8)	46.74 ± 1.90 ^a^	38.04 ± 1.81 ^b, c^	36.47 ± 3.40 ^c^	45.84 ± 3.15 ^a^	39.45 ± 0.73 ^b^	75.45 ± 0.19 ^a^	53.50 ± 2.86 ^c^	66.22 ± 0.62 ^b^	53.51 ± 2.08 ^c^	58.20 ± 3.03 ^b, c^
**β-phenyl ethanol** (1.0–1.88)	2.384 ± 0.044 ^a^	1.335 ± 0.472 ^a, b^	0.794 ± 0.098 ^b^	1.359 ± 0.382 ^a, b^	1.093 ± 0.456 ^a, b^	0.533 ± 0.004 ^a, b^	0.450 ± 0.043 ^a, b, c^	0.323 ± 0.030 ^b, c^	0.551 ± 0.179 ^a^	0.320 ± 0.034 ^c^
**Acetaldehyde** (0.02–0.12)	49.46 ± 1.42 ^a, b^	56.14 ± 4.99 ^a^	52.14 ± 3.28 ^a, b^	54.07 ± 8.72 ^a^	35.08 ± 2.43 ^c^	8.959 ± 1.216 ^d^	144.46 ± 13.96 ^a^	28.11 ± 0.53 ^c^	5.039 ± 0.536 ^d^	77.15 ± 21.80 ^b^

## Data Availability

The data presented in this study are available on request from the corresponding author.

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
