# Peer review of "Lentil Fortification and Non-Conventional Yeasts as Strategy to Enhance Functionality and Aroma Profile of Craft Beer"

_foods, 2022, doi:10.3390/foods11182787_

Round 1

Reviewer 1 Report

This study by Laura Canonico and colleagues investigated the effect of high-protein enriched craft beer: selected non-conventional yeasts in pure and mixed fermentations to enhance functionality and aroma profile of beer. This manuscript is logically clear. However, some concerns listed below need more explanations or modifications.

1. How do the author view the safety and reliability of using functional fermenting yeasts in probiotic beer? Are there any cases or laws and regulations in the world? There is a need to increase the discussion on the safety and application cases of probiotic yeast strains especially in human food. Therefore, it is necessary to reorganize the this manuscript on this point and make it clearly.

2. In section 3.4, the number of viable cell is c.a ≥6.5 log CFU/mL, which could not meet the requested number of viable probiotic bacteria (cell count c.a ≥7 log CFU/mL). Could the author give more explanation on this result?

3. Table 3, listing the thresholds of aroma flavors in the table would be more helpful for data analysis to draw conclusions. In addition, it is also easier for readers to understand the data.

4. In some sections, the results are not sufficient to support the conclusion. Discussion should be strengthened.

5. The full name of a microorganism should be written in the first mention in the main text.

Author Response

This study by Laura Canonico and colleagues investigated the effect of high-protein enriched craft beer: selected non-conventional yeasts in pure and mixed fermentations to enhance functionality and aroma profile of beer. This manuscript is logically clear. However, some concerns listed below need more explanations or modifications.

  1. How do the author view the safety and reliability of using functional fermenting yeasts in probiotic beer? Are there any cases or laws and regulations in the world? There is a need to increase the discussion on the safety and application cases of probiotic yeast strains especially in human food. Therefore, it is necessary to reorganize the this manuscript on this point and make it clearly.

Answer: The safety of the NCY used, their were undergone to a safety assessment protocol following the FAO-WHO guidelines for the valuation of probiotics in food used for GRAS microorganisms. The preliminary strain characterization  was reported in the reference 24 ( enclosed in the text) and further safety evaluation was subsequently carried out (unpublished data). Regarding the process reliability, do not show any technological trouble. We clarify this point in the section discussion  

  1. In section 3.4, the number of viable cell is c.a ≥6.5 log CFU/mL, which could not meet the requested number of viable probiotic bacteria (cell count c.a ≥7 log CFU/mL). Could the author give more explanation on this result?

Answer: The  number of viable probiotic of   7 log CFU/mL is request for bacteria, for probiotic yeasts,  to our knowledge, no limit has yet been established. However, the viability achieved here was quite satisfactory and close to that for bacteria

  1. Table 3, listing the thresholds of aroma flavors in the table would be more helpful for data analysis to draw conclusions. In addition, it is also easier for readers to understand the data.

Answer:   We added the OTV  in the first column of the table 3

  1.  

In some sections, the results are not sufficient to support the conclusion. Discussion should be strengthened.

Answer: The discussion was revised and improved

  1. The full name of a microorganism should be written in the first mention in the main text.

Answer: corrected in the text

Reviewer 2 Report

The manuscript explains the application of red lentil protein to the purification of craft beer and evaluated several functional properties. It was designed properly but need some corrections. L78-79: Please explain the relation between legumes, here lentil, and the function of craft beer. Firstly, the application of lentils in craft beer must be included in the title and the title is too long and should be corrected. The other comments are as follows:

L115: please explain briefly the protocol.

L137L: sensory evaluation by 10 panelists is weak and should increase the number of panelists.

Results: Please provide the GC plot along with the table.

Discussion must be included with the result and not provided as a separate section

Reviewer 3 Report

Canonico et al. reported that the NCY strains (L. thermotolerans and K. unispora) can enhance the functionality and aroma profile of craft beer. L. thermotolerans and K. unispora affected the amino acid profile of beers. Besides, L. thermotolerans tended to increase the contents of some higher alcohols, while K. unispora would promote the ethyl esters. The study is of great interest and application value. However, there still lies some small issues. Therefore, a minor revision was appropriate.

1.     Line13, the format of names of microbes. The first time of a species showed up should use full name not abbreviation. Furthermore, there were some format issues at line 56 (lactobacilli should be Lactobacilli), 57 (bifidobacterial should be Bifidobacterial), line 189 (K unispora), and line 272 (K unispora).

2.     Line 100-101, the mix ratios S. cerevisiae/ NYC yeast ratios were: 1:20 (L. thermotolerans), and 1:50 (K. unispora). Please explain why chose these ratios.

3.     Please use a rigorous subtitle for 3.4 part

4.     What is the meaning of word “idrolyzated”, and it seems a mistake? Please revise the whole manuscript to avoid this.

5.     Please revise the references’ part, line 446, 469

Author Response

Canonico et al. reported that the NCY strains (L. thermotolerans and K. unispora) can enhance the functionality and aroma profile of craft beer. L. thermotolerans and K. unispora affected the amino acid profile of beers. Besides, L. thermotolerans tended to increase the contents of some higher alcohols, while K. unispora would promote the ethyl esters. The study is of great interest and application value. However, there still lies some small issues. Therefore, a minor revision was appropriate.

  1. Line13, the format of names of microbes. The first time of a species showed up should use full name not abbreviation. Furthermore, there were some format issues at line 56 (lactobacilli should be Lactobacilli), 57 (bifidobacterial should be Bifidobacterial), line 189 (K unispora), and line 272 (K unispora).

Answer: corrected in the text.

  1. Line 100-101, the mix ratios S. cerevisiae/ NYC yeast ratios were: 1:20 (L. thermotolerans), and 1:50 (K. unispora). Please explain why chose these ratios.

Answer:  We set up the yeast ratios by the results of previous works  see references 23 and 36 of this manuscript

  1. Please use a rigorous subtitle for 3.4 part

Answer: corrected in the text.

  1. What is the meaning of word “idrolyzated”, and it seems a mistake? Please revise the whole manuscript to avoid this.

Answer: mistake  hydrolyzed is correct